# Influence of Nitrogen Fertilization Rate on Soil Respiration: A Study Using a Rapid Soil Respiration Assay

**Debankur Sanyal [1],\*** **, Johnathon Wolthuizen [2] and Anthony Bly [3]**

1   Department of Agronomy, Horticulture and Plant Science, South Dakota State University, Brookings, SD 57007, USA
2   Department of Agronomy, Horticulture and Plant Science, South Dakota State University Extension, Brookings, SD 57007, USA; johnathon.wolthuizen@sdstate.edu
3   Department of Agronomy, Horticulture and Plant Science, South Dakota State University Extension, Sioux Falls, SD 57106, USA; anthony.bly@sdstate.edu
\*   Correspondence: debankur.sanyal@sdstate.edu

**Abstract:** Efficient nitrogen (N) management is one of the primary objectives of agronomic research as N is expensive and a major environmental pollutant. Soil microbes regulate N cycling and soil respiration (SR) measures soil microbial activity. The Comprehensive Assessment of Soil Health (CASH) soil respiration protocol is a rapid test, and a study was designed to approve this test as a potential tool for corn (*Zea mays* L.) N management. Five locations were selected around South Dakota (SD) where corn received 0, 45, 90, and 180 kg N ha$^{-1}$ during summer of 2019. Soil samples were collected before planting and at the V6 corn growth stage to measure SR. We found that N fertilization increased SR and the highest SR was recorded at Ipswich (1.94 mg $CO_2$ g$^{-1}$) while SR was lowest at Bushnell (1.45 mg $CO_2$ g$^{-1}$). Higher SR was recorded at the sites where no-till farming was practiced, and soil had higher initial nitrate and organic matter content. SR was weakly correlated with corn grain yield, which indicated a potential area for future research. We concluded that split N application or an additional N application at a later growth stage might boost corn productivity in soil with higher microbial activity.

**Keywords:** soil respiration; nitrogen; corn; grain yield





## 1. Introduction

The essentiality of nitrogen (N) in sustaining soil fertility and crop production is well defined and many agronomic research have been directed toward finding the best N management for optimal crop yield [1–4]. The most popular way of supplementing the required amount of N to crops is through the addition of inorganic and/or organic N sources [5]. However, the trend is changing; with increasing financial and environmental stresses, understanding of nitrogen use efficiency (NUE) [6–8] and economic optimum nitrogen rate (EONR) [9–11] is gaining increased attention from crop producers. Therefore, much agronomic research in recent times has focused on designing an improved and more efficient N management strategy to enhance the potential of agricultural and food production systems.

Soil microbes play a major role in N cycling in soil. Fertilizer N can be lost into the atmosphere through gaseous forms of N, ammonia ($NH_3$), nitrous oxide ($N_2O$), and dinitrogen ($N_2$) or can leach down through the soil profile or runoff from the soil surface as nitrates ($NO_3^-$), nitrite ($NO_2$), and ammonium ($NH_4^+$) [12], and microbes play a crucial part in all of these transformations [13]. Microbial uptake of $NO_3^-$ and ammonium ($NH_4^+$) is an integral part of N dynamics in soil and, therefore, N addition should stimulate soil respiration (SR) [14]. Furthermore, N application rates could play a major role in shaping the microbial community structure and overall microbial activity. However, while the addition of N in soil has been reported to influence SR positively or negatively, many studies

have reported no influence too [15–17]. A high rate of N application ($25 \text{ g N m}^{-2} \text{ yr}^{-1}$) increased SR in a grassland system because of increased root and microbial biomass, and the ratio of $NO_3^-$ to $NH_4^+$, the mineralization rate, was significantly correlated with SR [18]. Additions of N in an N-limited soil was reported to stimulate an increase in SR initially, and with time created a C-limited state as N demand diminished [19].

Soil organic carbon (SOC) fractions have been used to determine optimum N rates for agricultural crop production and studies have found the labile carbon (C) fraction in soil to influence soil productivity [20–22]. C is essential for every life form on earth and soil microbes play a crucial role in mineralizing the most labile C fraction first and then relatively more stable C fractions in soil. Therefore, mineralization of C through the emission of carbon dioxide ($CO_2$) can be used to measure microbial activity in the soil [23,24]. SR is also referred to as C mineralization, which is a measure of $CO_2$ emission and commonly used as a soil health indicator for estimating microbial activity and active root growth on a global scale [18]. SR represents a major C flux between the soil and atmosphere [25]. As microbial population and activity increases in soil, the emission of $CO_2$, i.e., SR, also increases if not limited by available C [26].

Corn (*Zea mays* L.) is a very important crop in the US and had the highest acreage among crops grown in 2020 [27]. Therefore, managing N applications in corn production is one of the primary areas of research in US agriculture. This study was designed to understand the relationship between N-rate in corn and soil respiration (SR) rate as we hypothesized that SR might be a potential tool used to determine optimum N rate for corn. There are limited studies that have reported how soil respiration is related to corn grain yield or the yield of other cash crops that require N [28]. Therefore, the specific objectives of this study were i) to determine how soil respiration (microbial activity) was affected by N fertilization rates in a crop (corn) production system and ii) to report the relationship between soil respiration and crop yield under different N application rates.

## 2. Materials and Methods

### 2.1. Study Sites and Experimental Design

The experiment was laid out in corn fields at five different locations around South Dakota in 2019: Bushnell (44.331554, −96.652017), Garretson (43.652566, −96.482585), Howard (44.139415, −97.473298), Ipswich (45.4456786, −99.215861), and Mansfield (45.300989, −98.662870) (Figure 1). Except for the Bushnell site, which was conventionally tilled, all other study sites operated under no-till systems. Initial soil test information and additional site descriptions are given in Table 1. Four different N rates, 0, 45, 90, and 180 kg ha$^{-1}$ were applied to the corn plots before planting and each N rate was replicated four times at each location following a randomized complete block design (RCBD). Super-U (Koch™, Wichita, KS, USA), a slow-release urea fertilizer (46-0-0), was used as the N source that was broadcasted in the research plots. All plots had $4.6 \times 9.2 \text{ m}^2$ dimensions. Soil samples from the top 0–15 cm of each plot was randomly collected at the V6 corn growth stage for the soil respiration assay. Corn grain yields were recorded for the N rates.

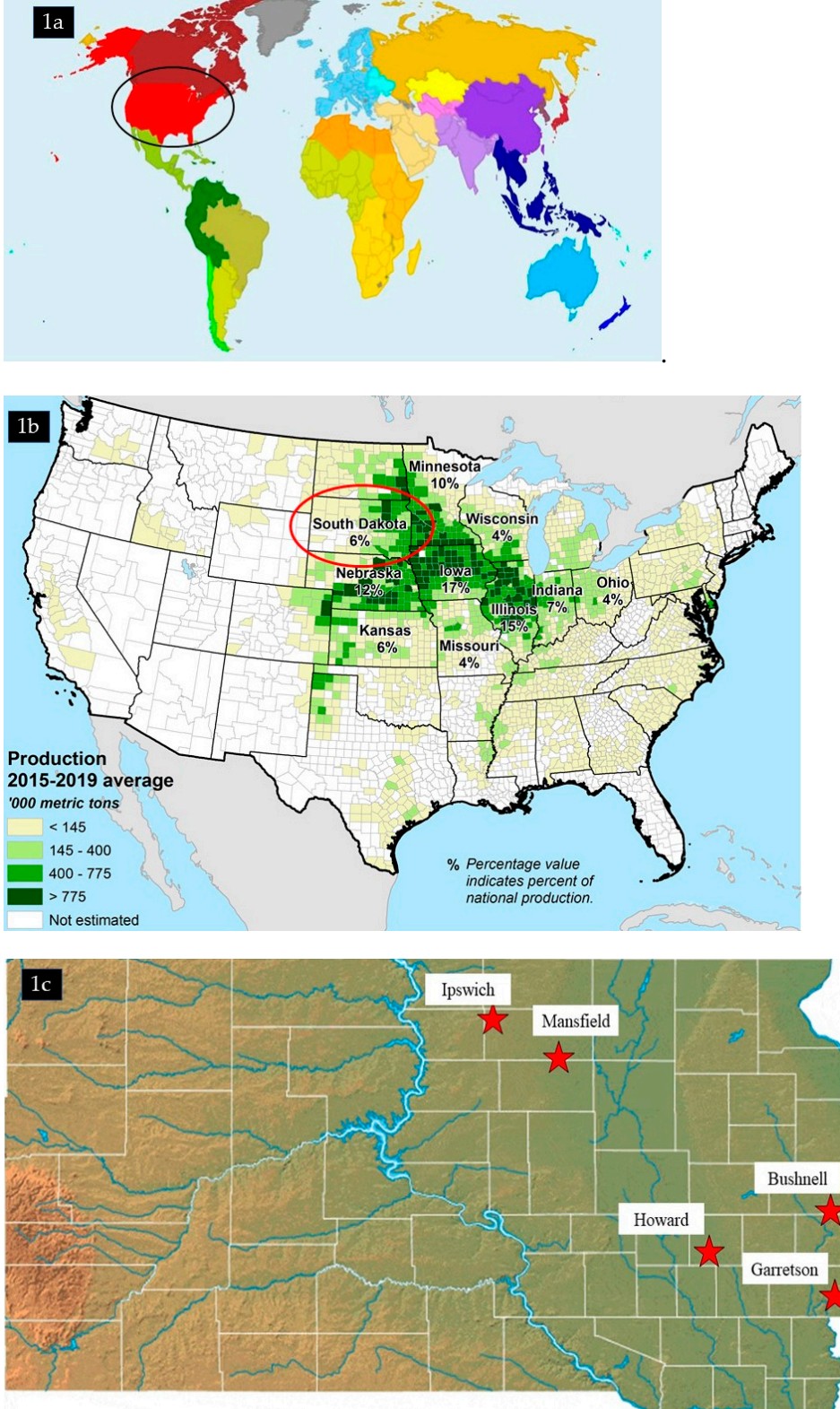

**Figure 1.** (**a**) World map with corn production zones (United State Department of Agriculture Foreign Agricultural Service); (**b**) United States Corn Production Map 2015–2019 (United State Department of Agriculture Foreign Agricultural Service); (**c**) topographical map of South Dakota showing the five study locations in 2019.

**Table 1.** Site descriptions and mean values of initial soil parameters in the corn plots at five study locations around South Dakota during the 2019 growing season.

| Parameters | Bushnell | Garretson | Howard | Ipswich | Mansfield |
|---|---|---|---|---|---|
| Tillage practice | Conventional | No-till | No-till | No-till | No-till |
| Grazing history | No | No | No | Yes | Yes |
| Total precipitation (mm, May–Sep) | 608 | 538 | 567 | 484 | 484 |
| Soil pH | 5.5 | 5.3 | 5.7 | 6.4 | 6.1 |
| Electrical conductivity (mS cm$^{-1}$) | 0.1 | 0.2 | 0.2 | 0.2 | 0.3 |
| Soil organic matter (%) | 3.3 | 3.7 | 5.5 | 5.3 | 4.0 |
| Soil nitrate (kg ha$^{-1}$, 0–60 cm) | 29 | 28 | 117 | 61 | 68 |
| Available phosphorus (ppm, 0–15 cm) | 13 | 12 | 12 | 24 | 6.3 |
| Available potassium (ppm 0–15 cm) | 94 | 120 | 172 | 273 | 261 |
| Soil sulfate (kg ha$^{-1}$, 0–60 cm) | 60 | 55 | 19 | 53 | 70 |
| Date of planting | 5/8/2019 | 5/18/2019 | 5/12/2019 | 5/14/2019 | 5/8/2019 |
| Date of fertilization | 5/3/19 | 5/16/2019 | 5/16/2019 | 5/3/2019 | 5/3/2019 |
| Date of soil (Corn V6 stage) sampling | 6/19/2019 | 6/25/2019 | 6/24/2019 | 6/18/2019 | 6/18/2019 |
| Date of harvest | 11/12/2019 | 11/08/2019 | Not harvested | 11/16/2019 | 11/16/2019 |

## 2.2. Incubation Study and Measuring Soil Respiration

Soil respiration was measured following a modified version of the Comprehension Assessment of Soil Health (CASH) protocol [29] where instead of a 4-day soil incubation, a 7-day soil incubation was performed to compare the efficiency of this methodology with the well-established modified Haney method [30]. Two sets of soil incubations were carried out following the two methods simultaneously to minimize error. Dried and ground (<0.2 cm) soil (20 g) was weighed on a perforated alumina tray and incubated inside glass mason jars (473 mL, Ball®, Rubbermaid Inc., Atlanta, GA, USA) at room temperature (~25 °C). Potassium hydroxide (KOH, Millipore Sigma, Germany; 0.5 M) and sodium hydroxide (NaOH, Fisher Scientific International L.L.C., Waltham, MA; 0.5 M) trap solutions were used for the CASH and Haney methods, respectively, and water was added to start the incubation. After seven days, the readings were recorded. Electrical conductivities of the trap solutions (KOH) were measured for the CASH analysis. NaOH was titrated against a 0.1 M hydrochloric acid (HCl) solution using phenolphthalein as the indicator (pink in basic solution and colorless in acidic solution) for the Haney method.

## 2.3. Statistical Analysis

Research data were analyzed using the R 4.0.2 software tool [31]. The data were checked for normality using the Shapiro–Wilk test [32] and then fitted into linear mixed models using the 'lmerTest' package [33]. In the mixed model [Equation (1)], replication was the random effect, while respiration, yield, and location were used as the fixed factors. Therefore, the model used in statistical analysis was defined as:

$$\text{N-rate} = \text{lmer(Respiration} \sim \text{N-rate*Location} + (1 \mid \text{Replications), data frame)} \qquad (1)$$

All the graphs were prepared in either Microsoft Excel or in R 4.0.2 using the 'ggboxplot' package.

## 3. Results and Discussion

### 3.1. Comparison between CASH Protocol and Haney Method

From the comparison, we found that the CASH and Haney methods correlate well ($R_2 = 0.98$) in estimating SR (Figure 2) and can be used as a standardized method in estimating SR; similar findings were reported by [23]. The CASH method is comparatively rapid and recent; therefore, we used this method to determine the impact of N rate on SR during corn growing season. In conclusion, the CASH method can be a potential choice to measure in other ecological studies, especially when SR is used as a soil health indicator.

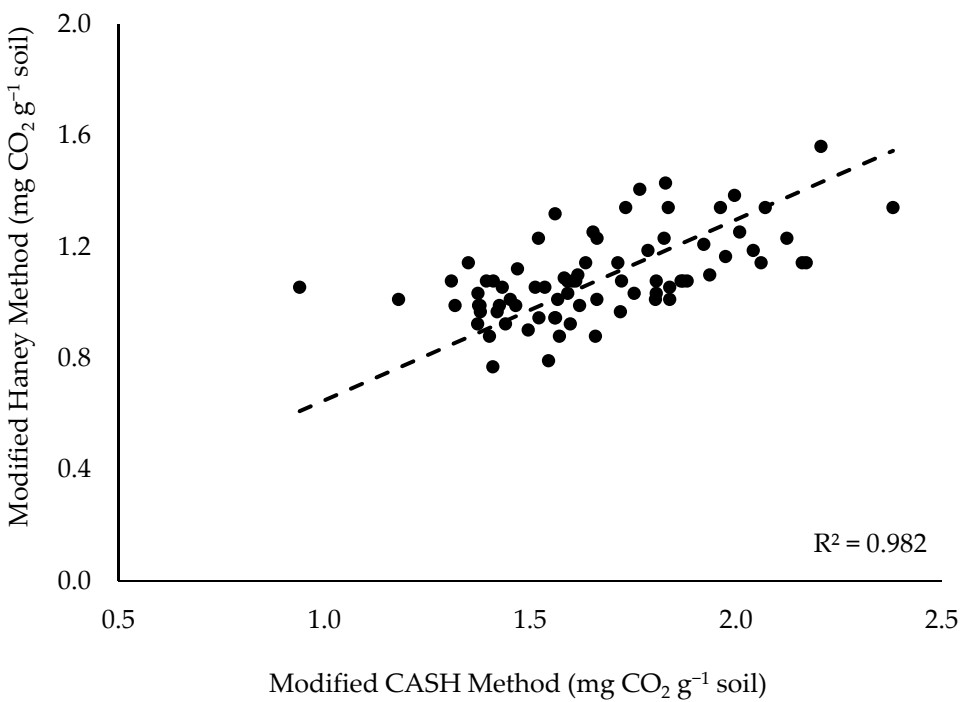

**Figure 2.** Comparison between modified CASH protocol [29] and modified Haney method [30] in estimating soil respiration.

*3.2. Impact of Nitrogen Fertilizer Application Rates on Soil Respiration*

Soil respiration increased with increasing N rates when averaged over the five locations in this study. The highest mean SR value (1.75 mg $CO_2$ $g^{-1}$ soil) was found under the highest N application rate (180 kg N $ha^{-1}$) followed by the 90 kg N $ha^{-1}$ rate, and these values were significantly higher than the 45 kg N $ha^{-1}$ and 0 kg N $ha^{-1}$ application rates (Figure 3). Similar outcomes were reported by [18,34–36] as they reported increases in SR with N fertilizer addition. However, this finding did not agree with several recent research conclusions that reported decreases in SR rate with N fertilizer applications in diverse agroecosystems ranging from forest to corn production [23,37–40]. Furthermore, [18] reported the $NO_3^-$ or $NH_4^+$ forms of N affected SR in Chinese grasslands while increasing SR through additions of N, but Ramirez et al. [39] reported that decrease in SR rates following N addition did not depend on the sources of N. Therefore, positive, negative, and neutral impacts of N rate on SR were found under diverse environmental conditions and it can be concluded that along with availability of N, other soil and environmental conditions dictate SR. Our research outcomes followed another recently published report by [23] as they reported increases in SR with additions of N fertilizers in one of their study sites was due to higher C storage in the soil. Additional inorganic N (fertilizer) in combination with high soil organic matter (SOM) content (Table 1) in soil enhanced SR in our study sites.

Overall, when the study sites were considered separately, SR rates did not significantly trend with N fertilization rates (Table 2), which also was reported by [23] where they reported 45 of their 49 study sites did not show significant differences in SR with N fertilizer additions. Similar conclusions were derived from our study as we did not find significant differences in SR among rates of fertilizer N when each site was considered separately. However, we found significant differences among SR values under different N fertilization rates. In addition, the mean SR values among the five study locations varied significantly (Table 2). The lowest mean SR (1.45 mg $CO_2$ $g^{-1}$ soil) was found at the Bushnell site, the only site that was conventionally tilled, and the highest mean SR value was found at Ipswich (1.94 mg $CO_2$ $g^{-1}$ soil) followed by Howard (1.80 mg $CO_2$ $g^{-1}$ soil) (Table 2). This outcome supported an earlier research article published by [41] who reported that minimal and no tillage resulted in higher SR compared to the conventionally

tilled sites. In this study, a potential contributing factor was high SOM and initial soil nitrate content in the soil as the highest SOM and soil nitrate status was found at Howard (5.5% and 117 kg N ha$^{-1}$) and Ipswich (5.3% and 67 kg N ha$^{-1}$). It was also noted that these sites practiced under no-till systems and, therefore, overall soil health (better soil biological health, more carbon and organic matter in the soil, etc.) was expected to be higher than at conventionally tilled sites.

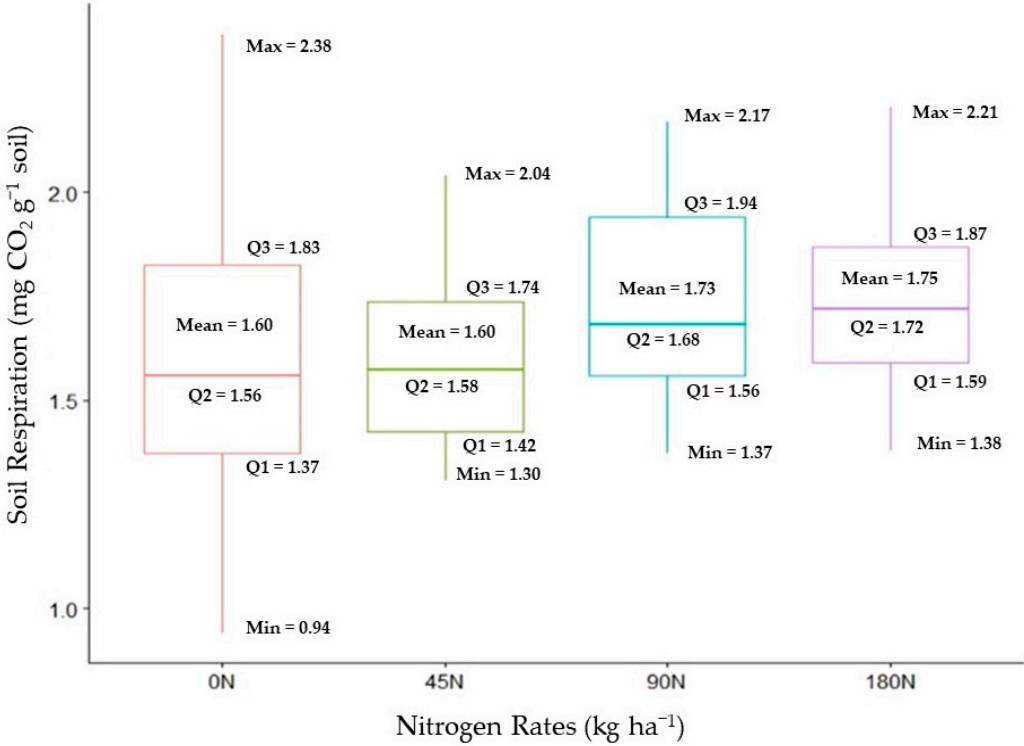

**Figure 3.** Mean soil respiration (mg $CO_2$ g$^{-1}$ soil) values at the V6 corn growth stage under four different nitrogen application rates, 0, 45, 90, and 180 kg N ha$^{-1}$, averaged over five locations in South Dakota during corn the growing season in 2019. In the figure, Mean = average values, Max = maximum values, Min = minimum values, Q1 = first quantile values, Q2 = median or second quantile values, and Q3 = third quantile values.

**Table 2.** Mean soil respiration (mg $CO_2$ g$^{-1}$ soil) values at the V6 corn growth stage under 0, 45, 90, and 180 kg N ha$^{-1}$ nitrogen fertilizer application rates at five locations, Aberdeen, Bushnell, Garretson, Howard, and Ipswich, around South Dakota in 2019.

| Locations | Nitrogen Rates | | | | |
|---|---|---|---|---|---|
| | 0 kg N ha$^{-1}$ | 45 kg N ha$^{-1}$ | 90 kg N ha$^{-1}$ | 180 kg N ha$^{-1}$ | Mean (Location) |
| Bushnell | 1.20 (0.17) | 1.52 (0.17) | 1.55 (0.12) | 1.54 (0.13) | 1.45D |
| Garretson | 1.43 (0.08) | 1.50 (0.08) | 1.58 (0.19) | 1.61 (0.09) | 1.53CD |
| Howard | 1.85 (0.37) | 1.65 (0.19) | 1.84 (0.16) | 1.85 (0.20) | 1.80B |
| Ipswich | 1.88 (0.22) | 1.85 (0.11) | 2.03 (0.10) | 1.99 (0.15) | 1.94A |
| Mansfield | 1.65 (0.14) | 1.49 (0.04) | 1.65 (0.08) | 1.76 (0.18) | 1.64C |
| Mean (N rate) | 1.60b | 1.60b | 1.73a | 1.75a | |
| *p*-values | Nitrogen rate | | | 0.01615 | |
| | Location | | | $1.745 \times 10^{-9}$ | |
| | Nitrogen rate × Location | | | 0.55176 | |

Nitrogen rate mean values followed by same lowercase letters are not significantly ($p < 0.05$) different; location mean values followed by same uppercase letters are not significantly ($p < 0.05$) different; standard deviation Values are given in the parentheses beside the mean values in the tables.

### 3.3. Corn Grain yield and Nitrogen Application Rate

Corn grain yield was significantly increased with higher N application rates (Table 3). The highest yield (10.7 mg ha$^{-1}$) was recorded under 180 kg N ha$^{-1}$ while the lowest yield (9.3 mg ha$^{-1}$) was recorded under the plots that did not receive any N fertilization. This finding confirmed previous reports of higher corn grain yield following increased N application rates [42,43]. Corn yields ranged from 11.9 to 12.9 mg ha$^{-1}$ at Mansfield (mean 12.4 mg ha$^{-1}$), from 10.9 to 12.7 mg ha$^{-1}$ at Bushnell (mean 11.5 mg ha$^{-1}$), from 9.61 to 10.5 mg ha$^{-1}$ at Garretson (10.0 mg ha$^{-1}$), and from 4.87 to 6.62 mg ha$^{-1}$ at Ipswich (5.8 mg ha$^{-1}$). The corn plots at Ipswich were affected by herbicide drift, therefore, the corn stand was poor and ultimately final yields were significantly lower than other locations. We also could not harvest the corn plots at Howard due to prolonged wet soil conditions later in the growing season (Table 1). Overall, we observed N fertilization responses from corn grain yields up to a rate of 180 kg N ha$^{-1}$. However, we did not have very high N fertilization rates, for example, over 250 kg N ha$^{-1}$, therefore, we cannot comment on economic optimum nitrogen rate (EONR).

**Table 3.** Mean corn yield (Mg ha$^{-1}$) values under four different nitrogen application rates, 0, 45, 90, and 180 kg N ha$^{-1}$, at five locations, Aberdeen, Bushnell, Garretson, Howard, and Ipswich, around South Dakota in 2019.

| Locations | Corn Yield (Mg ha$^{-1}$) | | | | |
|---|---|---|---|---|---|
| | 0 kg N ha$^{-1}$ | 45 kg N ha$^{-1}$ | 90 kg N ha$^{-1}$ | 180 kg N ha$^{-1}$ | Mean (Location) |
| Bushnell | 10.9 (1.4) | 11.3 (2.3) | 11.2 (1.2) | 12.7 (0.7) | 11.5A |
| Garretson | 9.61 (2.2) | 9.93 (2.2) | 10.1 (0.5) | 10.5 (1.4) | 10.0B |
| Ipswich | 4.87 (0.7)) | 5.18 (1.4) | 6.43 (0.8) | 6.62 (0.6) | 5.8C |
| Mansfield | 11.9 (0.8) | 12.2 (0.8) | 12.7 (0.2) | 12.9 (0.3) | 12.4A |
| Mean (N rate) | 9.31b | 9.67ab | 10.1ab | 10.7a | |
| *p*-values | N rate | | | 0.881 | |
| | Location | | | 1.868 × 10$^{-14}$ | |
| | N rate × Location | | | 0.991 | |

Nitrogen rate mean values followed by same lowercase letters are not significantly ($p < 0.05$) different, location mean values followed by same uppercase letters are not significantly ($p < 0.05$) different; standard deviation Values are given in the parentheses beside the mean values in the tables. P.S. Corn plots at the Howard, SD, site could not be harvested due to wet field conditions later in the 2019 growing season.

### 3.4. Soil Respiration Rate and Corn Grain yield

Our study revealed that SR at the V6 corn growth stage was negatively correlated with corn yield under all four N application rates, 0 kg N ha$^{-1}$ ($R^2 = -0.40$), 45 kg N ha$^{-1}$ ($R^2 = -0.09$), 90 kg N ha$^{-1}$ ($R^2 = -0.42$), and 180 kg N ha$^{-1}$ ($R^2 = -0.18$) (Figure 4a) when we included data from four sites (no yield data from Howard site), including Ipswich. Other published reports showing correlations between SR and corn yield or other crop yields are very limited. Sanyal et al. (2019) [28] reported that SR in cover cropped plots before corn planting was positively correlated with grain yield due to the additions of cover crop biomass that provided C for microbial activity and respiration. In this study, we measured SR at the V6 corn growth stage on plots where no cover crops were grown, therefore, the negative correlation was probably due to higher microbial activity that immobilized labile N and resulted in lower N availability to corn at later growth stages, thus lowering corn grain yield (Figure 3). We also discussed that lower yields in Ipswich were strongly influenced by herbicide drifts, and data from the Ipswich site might have influenced this negative correlation. When we excluded the data from the Ipswich site, we found positive correlations between SR at the V6 corn growth stage and corn yields (Figure 4b), which align with the report published by Sanyal et al. (2019) [28]. Therefore, we could not draw any definite conclusion on relationship between SR and corn yield, and more studies should be conducted to fully understand the relationship between SR and yields from crops that require N.

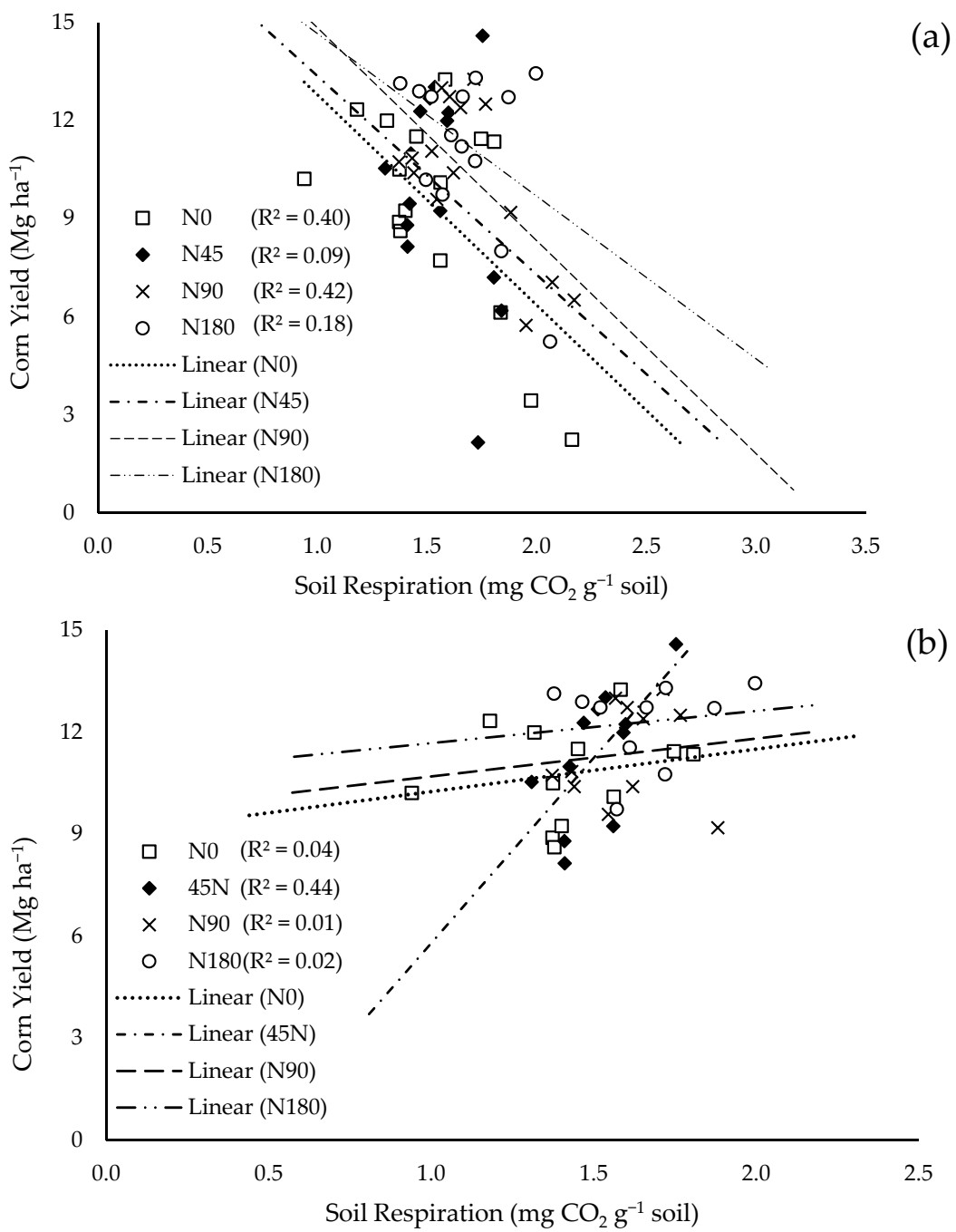

**Figure 4.** Correlations between soil respiration (mg $CO_2$ $g^{-1}$ soil) at the V6 corn growth stage and corn yield (Mg $ha^{-1}$) under four N application rates, 0, 45, 90, and 180 kg N $ha^{-1}$, in all the study sites combined (**a**) including Ipswich site and (**b**) excluding Ipswich site.

## 4. Conclusions

Our study established that the Comprehensive Assessment of Soil Health (CASH) soil respiration protocol can be used as a rapid method to estimate soil respiration. We also found that soil respiration is dependent on available nitrogen in soil and nonlimiting soil carbon and nitrogen. Soil respiration rate increased with inorganic nitrogen additions in the corn production systems. This study also revealed that soil respiration rate was correlated with corn yield, but this relationship was influenced by the study sites. An argument could arise from these conclusions that a split-nitrogen application or an additional nitrogen application might be necessary at the later corn growth stages to improve corn grain yield,

especially in soil with high microbial activity. This split-dose or additional nitrogen could help the crop to produce higher yields if microbes immobilize the labile nitrogen pool in the soil during critical reproductive growth stages. Therefore, nitrogen can be managed more efficiently and more economically if the soil microbial activity is known, but more research is required. In conclusion, this study opened a new avenue to investigate the relationship between soil respiration and economic returns such as grain yield from crop production systems.

**Author Contributions:** Conceptualization, D.S.; data curation, D.S.; formal analysis, D.S.; investigation, D.S. and J.W.; methodology, D.S.; project administration, D.S.; resources, A.B.; software, D.S.; supervision, D.S. and A.B.; validation, D.S., J.W. and A.B.; visualization, D.S.; writing—original draft, D.S.; writing—review and editing, D.S., J.W. and A.B. All authors have read and agreed to the published version of the manuscript.

**Funding:** This research received no external funding.

**Institutional Review Board Statement:** Not applicable.

**Informed Consent Statement:** Not applicable.

**Data Availability Statement:** Research data can be made available upon contacting the corresponding author at debankursanyal10@gmail.com.

**Acknowledgments:** The authors acknowledge Say Bway and Sandeep Kumar for their help and support in the analyses of samples.

**Conflicts of Interest:** The authors declare no conflict of interest.

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
