# Peer review of "Influence of Nitrogen Fertilization Rate on Soil Respiration: A Study Using a Rapid Soil Respiration Assay"

_nitrogen, doi:10.3390/nitrogen2020014_

Round 1

Reviewer 1 Report

The manuscript entitled “Influence of Nitrogen Fertilization Rate on Soil Respiration: A Study using a Rapid Soil Respiration Assay” presents a research related to the issue of efficient nitrogen management. 
In the Abstract, the authors refer to the CASH soil respiration protocol. The abbreviation must be explained in the place where the abbreviation was mentioned for the first time. The term was explained in lines 91-92 but currently, based only on the abstract, the readers are not able to understand clearly what do the authors mean. 
The authors have found that than nitrogen fertilization increases the soil respiration. It was also revealed that  higher soil respiration occurred at sites with no-tillage practice applied and in soils with higher content of nitrate and organic matter. 
Very important matter is that the study was designed to identify a relationship between nitrogen rate and soil respiration by the example of corn production. The idea was also to show the relationships between soil respiration and crop yield under different rates on nitrogen application. The directions of future research were also reasonably delineated. 
The concept of the research is logical and prepared in an understandable way, nevertheless some issues should be pointed out and amended:
-    Map on Figure 1 should be prepared with respect to top-down principle. The figure only shows the specific locations in South Dakota, with no regards to the overall view on the location on the background of the continent. 
-    In Figure 3, please correctly use the subscript and superscript in units of nitrogen rate and soil respiration. Please also clarify what data are presented in the Figure. Only mean values? What about the min, max, 25th and 75th percentiles or median? 
-    Lines 154-156: “This outcome supported an earlier report published by [41] who reported that minimal and no tillage had higher SR compared to the conventionally tilled sites” should be rewritten (“report” and “reported” in the same statement).
-    It should be clarified what are the values in brackets, presented in Table 2.
-    The discussion should be strengthened by more comprehensive comparison of the results obtained and outcomes of similar research presented in the scientific literature. 

Author Response

First of all, we thank you for your time and consideration. The suggestions were very constructive and will improve the impact of our manuscript. Please find the responses to the reviewer comments below:

-In the Abstract, the authors refer to the CASH soil respiration protocol. The abbreviation must be explained in the place where the abbreviation was mentioned for the first time. The term was explained in lines 91-92 but currently, based only on the abstract ,the readers are not able to understand clearly what do the authors mean.

Response: Thanks for the suggestion. I have abbreviated the term, please see line 14.

- Map on Figure 1 should be prepared with respect to top-down principle. The figure only shows the specific locations in South Dakota, with no regards to the overall view on the location on the background of the continent.

Response: Thanks for the suggestion. I have included the world map and a US map along with the map of South Dakota state with study locations. Please see figure 1a and 1b, lines 91-93.

- In Figure 3, please correctly use the subscript and superscript in units of nitrogen rate and soil respiration. Please also clarify what data are presented in the Figure. Only mean values? What about the min, max, 25th and 75th percentiles or median?

Response: Thanks for the correction and the suggestion. I have changed the superscripts and subscripts in the axis titles. Also, I have added mean, minimum, maximum, 25th and 75th percentiles and median values in the figure 3. Thanks, it was a great suggestion!

- Lines 154-156: “This outcome supported an earlier report published by [41] who reported that minimal and no tillage had higher SR compared to the conventionally tilled sites” should be rewritten (“report” and “reported” in the same statement).

Response: Thanks for the correction. The sentence is changed to “This outcome supported an earlier research article published by [41] who reported that minimal and no tillage had higher SR compared to the conventionally tilled sites.” Please see lines178-180.

- It should be clarified what are the values in brackets, presented in Table 2.

Response: Thanks for the suggestion! I have added a footnote to describe that there are standard deviation values in the brackets (parentheses) in both figure 2 and 3. Please see lines 167 and 205

- The discussion should be strengthened by more comprehensive comparison of the results obtained and outcomes of similar research presented in the scientific literature.

Response: Thanks for this constructive suggestion! We have added further explanations in the discussion. Please see the lines: 132-134, 149-151, 197-200.

Reviewer 2 Report

The topic of the manuscript “Influence of Nitrogen Fertilization Rate on Soil Respiration: A Study using a Rapid Soil Respiration Assay” is interesting and the authors had a good idea for a research project. The subject is relevant, the methodologies are adequate, and the volume of data seems to be enough for publication. I have no hesitation in recommending publication following minor revision.

General comments:

Abstract: Abstract presents summary, include key findings and the length of this part of the manuscript is appropriate.

Introduction: I consider that the structure of this section was well designed. Literature Review is adequate. Is effective, clear and well organized.

Material and methods: The methodology is well thought through.

Results and discussion: The results of the study are well presented, however, there is hardly discussion about Corn Grain yield and Nitrogen Application Rate. Authors should work on the discussion.

Conclusions: I believe that abbreviations should not be used in the Conclusion part.

The aim, range and results were clearly defined and demonstrate a good scientific knowledge of the nitrogen fertilization rate on soil respiration. The work contains appropriate testing methods and analyses of results.

Author Response

First of all, we thank you for your time and consideration. The suggestion was very constructive and will improve the impact of our manuscript. Please find the response to the reviewer comment below:

-The results of the study are well presented, however, there is hardly discussion about Corn Grain yield and Nitrogen Application Rate. Authors should work on the discussion.

Response: Thanks for the suggestion! We have added further explanations in the discussion. Please see the lines: 132-134, 149-151, 197-200.

Reviewer 3 Report

In the paper two methods to measure soil respiration (SR) are used in a maize experiment with 4 levels of N fertilization between 0 and 180 kg N ha-1. The methods are compared and it is tested if N level and yield level influences SR.

Remarks:

Materials and Methods

2.1 please give more information about the field trial:

  • form of N in fertilizer: nitrate, ammonium, urea?
  • Application broadcast or band placement?
  • Data of sewing, fertilization, V6 stage (=sampling of soil), harvest

2.2 line 95: 20g of soil: is that dried or as it comes out of the field? How is it made uniform for all locations?

2.3 statistical analysis

Line 107-109: what are you testing and what are declaring variables in the fixed model? For me as reader it might help if you write out the model(s).

3.1 Results and Discussion

Line 112-117 is (partly) a repetition of description in materials in methods. Can be left out here, and what is not in M&M can be added there.

3.2 line 160: how is “soil health” defined?

4 Conclusions

Line 204-205:

I don’t agree with this conclusion, the relation is too weak. It might be that there are not enough data to conclude.

The correlations are indeed very limited as you mentioned, especially the low R2 for N180 and N45.

If you look at the order of SR’s and corn yields:

The order in magnitude of yield from low to high is Ipsw-Garr-Bushn-Mansf

The order of SR for almost every Nlevel from high to low is Ipsw-Mansf-Garr-Bushn (except for N 45 where Garr and Bushn are switched). It looks like that sites are much more dominant in influence on SR than yield level since lowest and highest yield are on first and second place in the order of SR.

On top of that the lowest yield on Ipsw location seem to be very determining for the correlation. The low yield on this location, however, is caused by an external influence namely herbicide drift (line 175). Therefore it seems not correct to conclude that SR correlates with yield.

But maybe there is another way of reasoning possible? If so please explain that in the paper.

Line 206: the conclusion/suggestion for split N application: I don’t see on what results this is based. More explanation might help.

Author Response

Thanks you so much for your time, comments, and suggestions! We believe your comments improved our article and made it more attractive to the readers.

Please find our responses below in red font:

Materials and Methods

2.1 please give more information about the field trial:

form of N in fertilizer: nitrate, ammonium, urea?

Application broadcast or band placement?

Data of sewing, fertilization, V6 stage (=sampling of soil), harvest

Response: Thanks for these suggestions! We have added these information in the manuscript. Please see lines 83-84 for form of N and application method. Please see table 1, last 4 rows for the information on planting, fertilization, sampling and harvesting dates.

2.2 line 95: 20g of soil: is that dried or as it comes out of the field? How is it made uniform for all locations?

Response: This is a great question. We have added in the manuscript that we used dried and ground soil. Please see lines 111-112.

2.3 statistical analysis

Line 107-109: what are you testing and what are declaring variables in the fixed model? For me as reader it might help if you write out the model(s).

Response: This is a great comment! We have included the model in the manuscript for better explanation. Please see the lines 125-127.

3.1 Results and Discussion

Line 112-117 is (partly) a repetition of description in materials in methods. Can be left out here, and what is not in M&M can be added there.

Response: Thanks, it’s a great suggestion! We have edited the manuscript accordingly to avoid the repetition. Please see lines 132-137, 141-143.

3.2 line 160: how is “soil health” defined?

Response: It is a great input! We have added a brief explanation, please see the lines 211-212.

4 Conclusions

Line 204-205:

I don’t agree with this conclusion, the relation is too weak. It might be that there are not enough data to conclude.

The correlations are indeed very limited as you mentioned, especially the low R2 for N180 and N45.

If you look at the order of SR’s and corn yields:

The order in magnitude of yield from low to high is Ipsw-Garr-Bushn-Mansf

The order of SR for almost every Nlevel from high to low is Ipsw-Mansf-Garr-Bushn (except for N 45 where Garr and Bushn are switched). It looks like that sites are much more dominant in influence on SR than yield level since lowest and highest yield are on first and second place in the order of SR.

On top of that the lowest yield on Ipsw location seem to be very determining for the correlation. The low yield on this location, however, is caused by an external influence namely herbicide drift (line 175). Therefore it seems not correct to conclude that SR correlates with yield.

But maybe there is another way of reasoning possible? If so please explain that in the paper.

Response: That is a great comment! We appreciate it. We have added a new figure (figure 4b)  and a detailed explanation to clarify the results. The new figure depicts the scenario excluding the data from the Ipswich site. We believe this was a great suggestion and help the readers to understand our results better. Please see lines 238-239, 247-252, 257-258, 264-266.

Line 206: the conclusion/suggestion for split N application: I don’t see on what results this is based. More explanation might help.

Response: Thanks for the suggestion! We have added more explanation, please see lines 270-272.

Round 2

Reviewer 3 Report

The comments are well processed, in my view the paper has improvd.

I have no further comments.